# *Streptococcus salivarius* 24SMBc Genome Analysis Reveals New Biosynthetic Gene Clusters Involved in Antimicrobial Effects on *Streptococcus pneumoniae* and *Streptococcus pyogenes*

**DOI:** 10.3390/microorganisms10102042

**Published:** 2022-10-16

**Authors:** Gaia Vertillo Aluisio, Ambra Spitale, Luca Bonifacio, Grete Francesca Privitera, Aldo Stivala, Stefania Stefani, Maria Santagati

**Affiliations:** Medical Molecular Microbiology and Antibiotic Resistance Laboratory (MMARLab), Department of Biomedical and Biotechnological Sciences (BIOMETEC), University of Catania, 95123 Catania, Italy

**Keywords:** *S. salivarius* 24SMBc, BCGs, oral probiotic, WGS, bacteriocins

## Abstract

*Streptococcus salivarius* 24SMBc is an oral probiotic with antimicrobial activity against the otopathogens *Streptococcus pyogenes* and *Streptococcus pneumoniae*. Clinical studies have reinforced its role in reducing the recurrence of upper respiratory tract infections (URTIs) and rebalancing the nasal microbiota. In this study, for the first time, we characterized 24SMBc by whole genome sequencing and annotation; likewise, its antagonistic activity vs. *Streptococcus pneumoniae* and *Streptococcus pyogenes* was evaluated by in vitro co-aggregation and competitive adherence tests. The genome of 24SMBc comprises 2,131,204 bps with 1933 coding sequences (CDS), 44 tRNA, and six rRNA genes and it is categorized in 319 metabolic subsystems. Genome mining by BAGEL and antiSMASH tools predicted three novel biosynthetic gene clusters (BGCs): (i) a Blp class-IIc bacteriocin biosynthetic cluster, identifying two bacteriocins *blpU* and *blpK*; (ii) an ABC-type bacteriocin transporter; and (iii) a Type 3PKS (Polyketide synthase) involved in the mevalonate pathway for the isoprenoid biosynthetic process. Further analyses detected two additional genes for class-IIb bacteriocins and 24 putative adhesins and aggregation factors. Finally, in vitro assays of 24SMBc showed significant anti-adhesion and co-aggregation effects against *Streptococcus pneumoniae* strains, whereas it did not act as strongly against *Streptococcus pyogenes*. In conclusion, we identified a novel *blpU*-*K* bacteriocin-encoding BGC and two class-IIb bacteriocins involved in the activity against *Streptococcus pneumoniae* and *Streptococcus pyogenes*; likewise the type 3PKS pathway could have beneficial effects for the host including antimicrobial activity. Furthermore, the presence of adhesins and aggregation factors might be involved in the marked in vitro activity of co-aggregation with pathogens and competitive adherence, showing an additional antibacterial activity not solely related to metabolite production. These findings corroborate the antimicrobial activity of 24SMBc, especially against *Streptococcus pneumoniae* belonging to different serotypes, and further consolidate the use of this strain in URTIs in clinical settings.

## 1. Introduction

Over the last few years, increasing data on the microbiome and its theater of activity (structural elements, metabolites, environmental conditions) has highlighted the essential role of commensal bacteria in the development of the human body and its physiology [1], establishing the idea that human health also depends on the symbiotic relationship with a “healthy” microbiome and its crucial role in preventing infection due to its bacterial competition ability for the same niche, nutrients, and adhesion surfaces [2]. Although antibiotic therapy is the standard treatment for bacterial infections, the indiscriminate use of antibiotics in many human endeavors is one of the principal reasons why microbial resistance is increasing at an alarming rate worldwide [3]. Moreover, another critical implication of antibiotic use is how it affects not only pathogens but also commensal bacteria. This might alter the dynamic equilibrium between the human body and its microbiome, leading to a diversity loss described as “dysbiosis”, which can give an advantage for the emergence and outbreak of pathogens [4].

In particular, the upper respiratory tract (URT) offers several niches that are normally colonized by different microbial communities, acting as gatekeepers to respiratory health [2]. Infections of the upper respiratory tract (URTIs) are very common in both children and adults and include the common cold, laryngitis, pharyngitis/tonsillitis, acute rhinitis, acute rhinosinusitis, and acute otitis media. URTIs are usually one of the main reasons for seeking medical care and for inappropriate antibiotic prescriptions, with potential consequences in the prevalence of antibiotic resistance [5,6]. Although the majority of URTIs are usually caused by viruses, bacteria can also be the causative agent, often leading to clinical challenges related to a higher morbidity and a chronic progression of the disease [7]. Furthermore, studies of the nasopharyngeal microbiome demonstrated the important role of the microbiota in URTIs because microbial imbalance, often caused by antibiotic therapy in recurring infection, can determine the reduction or loss of beneficial commensal bacteria that exert a protective role, averting the host colonization and the development of bacterial infections [2]. Thus, it is important to focus on alternative strategies to prevent and treat infection in order to reduce antibiotic use and the incidence of drug resistance [8]. Local bacteriotherapy has been proposed as a potential strategy, based on the administration of oral probiotics that can exert beneficial effects to maintain and/or restore healthy microbial communities within the URT and reduce the severity, duration, and incidence of re-infection, interfering with the adhesion and colonization of potential pathogens by the production of antimicrobial peptides (bacteriocins) and immunomodulatory mediators; likewise, competitive adherence and co-aggregation between pathogens and probiotic strains can enhance the antagonistic activity of probiotics [9].

Evidence shows the important role played by α-streptococci, notably those with antagonistic activity against pathogens, in the protection from URTIs. Numerous studies have reported how a significant reduction in α-streptococci in the URT is correlated with a higher incidence of re-infections in patients with streptococcal pharyngotonsillitis, but also with a higher probability of otitis media in children [10].

The first oral streptococcal probiotic used in clinical setting was *S. salivarius* K12, the producer of plasmid-carried salA/B bacteriocins, in the treatment of halitosis and *S. pyogenes* infections. *S. salivarius* K12 and M18, already commercially available, also reduced recurrent streptococcal pharyngitis, whereas M18 reduced dental plaque scores and *S. mutans* count [11]. Additionally, *Streptococcus*
*salivarius* 24SMBc is an oral probiotic that, when administered intranasally, reduced the risk of new episodes of acute otitis media in otitis-prone children [12]. Further studies have reported the potential synergic benefits of using an oral probiotic formulation, already commercially available, of *S. salivarius* 24SMBc and *S. oralis* 89a (98:2 ratio) for its ability to rebalance the nasal microbiota improving the health status as well as to inhibit the biofilm formation of URT pathogens and prevent recurring URTIs in infant and adult populations [13,14,15,16,17].

In particular, the antimicrobial activity of *S. salivarius* 24SMBc has been well-characterized for its ability to adhere to epithelial cells and for *blpU* bacteriocin-production, mainly targeting other pathogenic streptococcal strains, namely *Streptococcus pyogenes* and *Streptococcus pneumoniae* [18,19]. In fact, oral streptococci are among the major producers of bacteriocins (i.e., ribosomal synthesized and post-translationally modified peptides (RiPP) [20]) that, like non-ribosomal peptides (NRP) and polyketides (PK), belong to natural bioactive products (NPs). These compounds often have antimicrobial activity, although they are highly diverse in their structure [21].

The production of natural bioactive products (NPs) is usually controlled by genes for biosynthetic regulation, transport, and immunity, which are physically clustered together on the genome to form biosynthetic gene clusters (BGCs). Thus, genome mining to identify potential BGCs using whole genome sequencing has become a pivotal element to discover novel NP biosynthesis pathways of interest, since many NPs could have potential therapeutic applications [22].

In light of these considerations, in this study, we report, for the first time, the genomic characterization of *S. salivarius* 24SMBc combined with genome mining tools to obtain new insights into its antimicrobial effects; furthermore, its antimicrobial activity was evaluated by in vitro co-aggregation and competitive adherence tests against different *S. pneumoniae* and *S. pyogenes* serotypes.

## 2. Materials and Methods

### 2.1. Bacterial Strains and Growth Conditions

*S.**salivarius* 24SMBc isolated from the nasopharyngeal swab of healthy children [18] and the indicator strains used in this study were grown in Tryptic Soy Broth (TSB) and Tryptic Soy Agar (TSA) (Oxoid, Basingstoke, UK) with 5% horse blood (Thermo Scientific, Basingstoke, UK). Cultures were incubated overnight at 37 °C in 5% CO_2_. Stock cultures of all strains were stored in TSB with 20% glycerol at −80 °C. The indicator strains were selected from our microbial bank at the MMARLab and comprised four *S. pneumoniae* belonging to serotype 19A (*S. pneumoniae* BT), serotype 15C (*S. pneumoniae* C2), serotype 9V (*S. pneumoniae* A3), and one non-typeable (*S. pneumoniae* M4); three *S. pyogenes,* two belonging to serotype M1, *S. pyogenes* MGAS 5005 (ATCC BAA-947) and *S. pyogenes* Spy3537 [23], and one to serotype M18 (*S. pyogenes* 2812A) [24].

### 2.2. Whole-Genome Sequencing, De Novo Genome Assembly

Genomic DNA isolated from *S. salivarius* 24SMBc was extracted using a PureLink™ Genomic DNA Mini Kit (Thermo Fisher Scientific, Waltham, MA, USA) according to the manufacturer’s instructions. Genomic DNA was quantified by a Qubit 2.0 fluorometer (dsDNA HS assay, Invitrogen, Waltham, MA, USA) and its purity grade was assessed by a Nanodrop2000 Spectrophotometer (Thermo Fisher Scientific, Waltham, MA, USA), evaluating the DNA absorbance ratios at 260 nm/280 nm and 260 nm/230 nm and obtaining a total DNA concentration of 1.4 µg/µL with the absorbance ratio 260/280 of 1.9 and 260/230 of 2.15. DNA integrity was also confirmed by the agarose gel electrophoretic assay. Samples were loaded into a 0.8% agarose gel and ran for 3 h at 70 V in 0.5 × Tris-Borate-EDTA buffer (Appendix A). Genomic DNA (60 ng total) was used to prepare libraries using the Nextera XT DNA Library Prep Kit (Illumina, San Diego, CA, USA) and sequenced by the Illumina MiSeq platform [25] following the manufacturer’s protocol. The raw data were provided in paired end reads (2 × 250 Read Length), resulting in a 258 × coverage. Sequence reads were quality controlled using Trimmomatic [26] and genome assembled by SPAdes v3.14.0 through the MEGA annotator pipeline [27].

### 2.3. Genomic Data Accession Number

The *S. salivarius* 24SMBc genomic reads were deposited in the National Center for Biotechnology Information (NCBI) SRA database under SRR20769826, the BioProject accession number PRJNA865118.

### 2.4. Genome Annotation

The genome was annotated with Prokka v. 1.13 [28], ribosomal RNA gene predictions were performed with RNAmmer v1.2 [29], while transfer RNA gene predictions were achieved with tRNAscan-SE v1.21 [30]. Rapid annotation using subsystem technology (RAST) was applied for microbial genome annotation to divide the genome into different subsystems, which are defined as “a set of functional roles” connected to specific genes of the available annotated genome [31].

Prokka output (Appendix A) was also used as the input for SPAAN [32], a software for the prediction of adhesins and adhesin-like proteins; protein domains were further analyzed using the NCBI’s Conserved Domain Database (CDD) with the BatchCD tool (https://www.ncbi.nlm.nih.gov/Structure/bwrpsb/bwrpsb.cgi accessed on 1 April 2022) to confirm the detection of bacterial adhesins. PathogenFinder of the CGE server and MP3 (Prediction of Pathogenic Proteins) were used to predict the potential pathogenicity [33,34].

### 2.5. In Silico Prediction and Characterization of Bacteriocin Biosynthetic Gene Clusters

Microbial genome mining was performed using the antiSMASH 6.0 pipeline (antibiotic and Secondary Metabolites Shell) to predict classic and novel biosynthetic gene clusters (BGCs) coding for secondary metabolites such as polyketide synthetases (PKS) and post-translationally modified peptides (RiPPs) [35]. The BAGEL4 tool (BActeriocin GEnome minimal tooL) was used to identify ribosomal synthesized and post-translationally modified peptides (RiPPs) and cluster genes related to bacteriocin synthesis [36]. The amino acid sequences of the core-peptides identified within the BGCs were further analyzed by the CD-search tool.

### 2.6. Adhesion Assay

For adhesion assays, the HEp-2 cell line (ATCC CCL-23™) was used. The cells were maintained in Dulbecco’s modified Eagle medium (D-MEM) (Sigma-Aldrich, Saint Louis, MO, USA), supplemented with 2 mM L-glutamine, antibiotics (penicillin 100 U/mL; streptomycin 100 U/mL; amphotericin B 0.25 µg/mL) and 6% (*v*/*v*) fetal bovine serum (FBS) (Thermo Fisher Scientific, Waltham, MA, USA). Confluent monolayers were dissociated by incubation with 0.25% trypsin/EDTA (0.25% (*w*/*v*) trypsin, 0.1 mM EDTA) (Life Technologies, Carlsbad, CA, USA) for 10 min at 37 °C in 5% CO_2_, then seeded into 24-well plates at a concentration of 1.5 × 10^5^ cells/well in D-MEM supplemented with 2% FBS for adhesion assays. Before the tests, cells were washed at least twice with 500 µL of saline phosphate buffer (PBS) (Sigma-Aldrich, Saint Louis, MO, USA), then 500 μL of DMEM was added without antibiotics and with 2% FBS. To determine the adherence degree of each strain, 10 µL of bacterial culture containing 1.5 × 10^8^ CFU/mL was added to the HEp-2 cells. PBS was used as a negative control and 100 U/mL heparin was used as a positive control to block pneumococcal adhesion. After 1 h incubation, unbound bacteria were washed twice with PBS 1X (Gibco™, Thermo Fisher Scientific, Waltham, MA, USA), and the cells and adhering bacteria were removed by trypsin/EDTA (0.25%). The viable bacterial counts (CFU/mL) were carried out on TSA with 5% horse blood after overnight incubation [37].

### 2.7. Anti-Adhesion Activity of 24SMBc against S. pneumoniae and S. pyogenes on HEp-2 Cells

All strains were inoculated in 20 mL of Todd Hewitt broth (THB) (Oxoid, Basingstoke, UK), supplemented with 0.5% (*w*/*v*) yeast extract (THY) (Oxoid, Basingstoke, UK) and incubated at 37 °C in 5% CO_2_ for 16 h. The competitive exclusion assays were performed as described by Dunne et al. with some modifications [38]. A total of 100 µL of overnight bacterial culture was inoculated in 20 mL of THY to reach mid-log phase (approximately 3 h for *S. salivarius* 24SMBc and *S. pyogenes* and 4 h for *S. pneumoniae*). After centrifugation, the pellet was resuspended in 0.85% (*w*/*v*) NaCl (Merck, Darmstadt, Germany) to obtain the concentration of 1.5 × 10^9^ CFU/mL for *S. salivarius* 24SMBc (optical density OD_600_ ≈ 0.30) and 1.5 × 10^8^ CFU/mL for the *S. pyogenes* (OD_600_ ≈ 0.30) and *S. pneumoniae* strains (OD_600_ ≈ 0.35). To assess the anti-adhesion activity by exclusion test, 10 µL per well of 24SMBc (10^9^ CFU/mL) was administered to cells 1 h prior to the addition of *S. pneumoniae* or *S. pyogenes* strains on the 24-well plate. After incubation, cells were washed with PBS 1X twice and adherent bacteria were quantified as aforementioned. The viable bacterial count, and consequently the detection of streptococcal strains, was obtained by different phenotypical features and morphology on TSA with 5% horse blood. The interference of probiotic *S. salivarius* 24SMBc versus streptococcal pathogens was expressed as the percentage of adhering bacteria normalized to *S. pneumoniae* and *S. pyogenes* adhesion when alone.

The statistical analysis was performed by GraphPad Prism 6 software (GraphPad software Inc., San Diego, CA, USA). These experiments were performed in triplicate and the statistical analysis of adherence interference was performed by the multiple *t*-test, *p* value: * *p* < 0.05, ** *p* < 0.01, and *** *p* < 0.001.

### 2.8. LDH Assay for Cytotoxicity

To evaluate the cytotoxic effect of *S. salivarius* 24SMBc on HEp-2, lactate dehydrogenase (LDH) release was measured. After 3 h of incubation with *S. salivarius* 24SMBc, the supernatants from HEp-2 monolayers grown in 24-well plates were collected. The levels of LDH in the supernatants were assayed in triplicate using a colorimetric Cytotoxicity Detection Kit (Roche, Mannheim, Germany) according to the manufacturer’s instructions. LDH is a stable cytosolic enzyme of eukaryotic cells, an indicator of necrotic cell death when released [39]. HEp-2 cells exposed to Triton X-100 (0.9%) were used as a control of the total release (100% LDH release). The background level (0% LDH release) was determined with a bacteria-free culture medium. One-way ANOVA was used for the analysis of the LDH assay, *p* value: * *p* < 0.05, ** *p* < 0.01, and *** *p* < 0.001.

### 2.9. Auto- and Co-Aggregation Assay

The auto- and co-aggregation tests were conducted following the protocol of Chaffanel et al. with a few modifications [40]. Each strain was grown in THY to reach log phase, as described previously. Pellets were centrifuged at 3000× *g* for 15 min at 4 °C and resuspended in a peptone water solution (casein peptone 0.1% (p/v) (Sigma-Aldrich, Saint Louis, MO, USA) to obtain bacterial suspensions with an optical density of OD_600_ ≈ 0.8. To test for the co-aggregation levels, suspensions of *S. salivarius* 24SMBc were mixed with an equal volume of indicator strain suspensions, then incubated at room temperature for 1 h. Both the bacterial suspensions alone for the auto-aggregation measurements, and together in co-aggregation were placed in triplicate in a 96-well plate (100 μL per well) for a T0 reading with the microplate reader SYNERGY. After 1 h of incubation at room temperature, centrifugation at 650× *g* for 2 min was performed to precipitate non-aggregated bacteria, then a second reading at T1 was taken by placing 100 μL of the supernatant of the bacterial suspensions in the microtiter.

The auto-aggregation percentage is expressed as 1 − (*AT*1/*AT*0) × 100, and the percentage of co-aggregation (CoA%) was calculated as (*AT*0 − *AT*1/*AT*1) × 100. The co-aggregation values of *S. salivarius* 24SMBc with the indicator strains were considered significant when the co-aggregation percentage was higher than the auto-aggregation percentage of each pathogenic strain. Co-aggregation assays were analyzed using ANOVA with Fisher’s significant difference (LSD) test, *p* value: * *p* < 0.05, ** *p* < 0.01, and *** *p* < 0.001. All experiments were performed in triplicate.

## 3. Results

### 3.1. Genome Annotation

The total 2,131,204 bp size of the 24SMBc genome was assembled in one chromosome with 39.85% GC content composited in 32 contigs with an N50 value of 171,173. The genome carried 1933 coding genes (CDSs), 44 tRNA, and six rRNA (Table 1). Genome analysis using PathogenFinder and MP3 tools confirmed that 24SMBc is a non-pathogenic strain as non-pathogenic protein families were identified.

The RAST annotation server detected 217 subsystems distributed in twenty-five categories and subcategories including, mainly, 226 predicted genes related to “amino acids and derivates”, 164 to “carbohydrates”, 193 to “protein metabolisms”, 114 to “cell wall and capsule”, and 111 to “RNA metabolisms”. Interesting subcategories comprised the mevalonate branch of isoprenoid biosynthesis and isoprenoids for quinones in the “fatty acids, lipid and isoprenoids” category; and cobalt-zinc-cadmium resistance, copper homeostasis, mercuric reductase, and the mercury resistance operon in the “resistance to antibiotics and toxic compounds” section. Moreover, no putative genes were identified either as determinants of virulence or as genes associated with mobile elements (Figure 1).

Prokka annotation (Appendix A) also detected four genes involved in bacteriocin production: two bacteriocin class IIc, *blpU* of 59 amino acids (genome locus tag LFGOLEEL_01643) and *blpK* of 64 amino acids (genome locus tag LFGOLEEL_01651), and two additional class IIb bacteriocins of the bact_IIb_cerein family consisting of 52 and 55 amino acids (genome locus tag LFGOLEEL_01810 and LFGOLEEL_01811) (Table 2).

Through a combination of software-assisted (SPAAN), CD-search tool, and manual annotation, a total of 24 putative adhesins and/or aggregation substances were identified in sixteen loci on the 24SMBc genome (Table 3) and class A and B sortase (*srtA*, locus tag LFGOLEEL_00109 and *srtB* locus tag LFGOLEEL_ 01150). Fifteen of them displayed a typical cell-wall signature with LPxTG motif, which is recognized by the sortase enzyme known to be involved in the surface anchorage, while six showed an YSIRK_motif in their signal peptide and three a KxYKxGKxW motif.

### 3.2. Microbial Genome Mining for Biosynthetic Gene Clusters

Genome mining for bacteriocin biosynthetic gene clusters by BAGEL4 identified one bacteriocin biosynthetic gene cluster (BBCG) in contig 24 (24.23.AOI_01; 77377-106176) with two core peptides: BlpU (pfam 10439), associated with the *blp*-cassette (accession number KY347796) as previously described [17], and BlpK (pfam10439) downstream of the cassette; moreover, this BBGC contained an additional protein, BmbF, a putative dual-specificity RNA methyltransferase RlmN (PF04055) (Figure 2).

Through antiSMASH v.6.0, a total of three putative biosynthetic gene clusters (BGCs) were detected, namely, two bacteriocin clusters (BBCG1 and BBCG3) and another cluster BCG2, labeled as a type III polyketide synthase cluster (T3PKS) (Appendix A). In particular, the bacteriocin cluster BBGC1 (contig_6 RiPP-like) contained the peptidase_C39 (PF03412) as a core peptide, an ABC-type bacteriocin transporter that is a signature for the salivaricin biosynthetic gene cluster of *Lactobacillus salivarius* (accession number EF592482.1; MIBiG accession number BGC0000624) (Appendix A).

The Blp bacteriocin cluster previously described by BAGEL was identified in BBCG3 (Contig_24_ RiPP-like) by antiSMASH, showing identical genetic organization including two bacteriocin_IIc, processing/transport and self-immunity proteins (Appendix A). Interestingly, the antiSMASH cluster blast that showed ten genetic clusters among those most similar to BBCG3 highlighted a different gene organization of this cluster, typical with each *S. salivarius* strains while maintaining the main functional genes, thus BBCG3 appeared to be a specific strain locus (Appendix A). BCG2 was detected to encode the multidomain enzyme type III polyketides synthase (PKs) (Contig_11 T3PKS) that, through the mevalonate pathway, is involved in the isoprenoid biosynthetic process. BCG2 presented a signature for viguiepinol biosynthetic gene cluster from *Streptomyces* sp. KO-3988 (accession number AB183750.1, MIBiG accession number BGC0000286) and showed a core biosynthetic gene encoding for hydroxymethylglutaryl-CoA synthase (HMG_CoA_synt_N, PF01154; HMG_CoA_synt_C, PF08540), which catalyzes the condensation of acetyl-CoA with acetoacetyl-CoA to produce HMG-CoA and CoA, and four additional biosynthetic genes encoding for (i) nucleotidyl transferase (NTP_transferase (PF00483); (ii) two glycosyl transferase group 1 (Glycos_transf_1, PF00534); and (iii) an alpha/beta hydrolase (Abhydrolase_1, pfam00561) (Appendix A).

### 3.3. Antagonistic Activity of S. salivarius 24SMBc against S. pneumoniae and S. pyogenes Adherence to HEp2 Cells

The antibacterial effect of *S. salivarius* 24SMBc vs. both *S. pneumoniae* group serotypes 19A, 15C, 9V, and NT and *S.*
*pyogenes* group serotypes M1 and M18 was evaluated by the competitive exclusion test on HEp-2 cell lines. Before performing the interference tests, the adherence properties of each streptococcus on HEp-2 cells were evaluated as they could vary considerably between different streptococcal strains. After 1 h of incubation, *S. salivarius* 24SMBc achieved a viable count of 7.10 × 10^8^ CFU/mL as HEp-2 adherent bacteria, whereas for *S. pneumoniae* strains, the adherence rates were approximately 2.24 × 10^5^ CFU/mL except for *S. pneumoniae* A3, showing 4.77 × 10^4^ CFU/mL, and the *S. pyogenes* group reached a viable count of 3.57 × 10^6^ CFU/mL. The results shown in Figure 3A are expressed as the percentage of adherence normalized to the number of adherent streptococci alone. 24SMBc inhibited the adhesion of the *S. pneumoniae* group with statistically significant differences, in particular, the *S. pneumoniae* serotype 19A and 15C (*S. pneumoniae* BT and *S. pneumoniae* C2), showing a reduction in the adhesion of 90% and 67%, respectively, as reported in Figure 3A (Appendix A). Otherwise, *S. pyogenes* strains showed a different behavior, maintaining a level of adhesion of approximately 10^6^ CFU/mL when incubated with 24SMBc. As a result, the reduction in the percentage of adherence was not statistically significant (Figure 3B).

### 3.4. Cytotoxic Effect of S. salivarius 24SMBc on HEp-2

*S. salivarius* 24SMBc had no cytotoxic effects on HEp-2 cells by measuring the LDH release from cells incubated with the bacteria for up to 3 h. The results showed that the amount of LDH release in the supernatant of HEp-2 cultured in the absence or presence of the bacteria was unchanged, indicating that *S. salivarius* 24SMBc at 10^9^ CFU/mL did not cause cell lysis (Figure 4).

### 3.5. Auto-Aggregation and Co-Aggregation Ability

Aggregation properties were assayed with the auto-aggregation and co-aggregation tests measuring two different characteristics of the strains. The auto-aggregation rate of *S. salivarius* 24SMBc measured after 1 h of incubation exhibited the following value of 81.8 ± 0.013%. *S. pyogenes* strains (*S. pyogenes* 2812A, *S. pyogenes* 35370, and *S. pyogenes* 5005), despite having a strong selective interaction value with *S. salivarius* 24SMBc, possessed strong self-aggregation properties (83.6%, 73.7%, and 73.1%, respectively), however, this was not statistically significant. On the other hand, *S. salivarius* 24SMBc exhibited a significant co-aggregation rate with *S. pneumoniae* strains belonging to different serotypes, with values ranging between 67 ± 0.003% and 52.5 ± 0.015%, as illustrated in Figure 5.

## 4. Discussion

In prior studies, *S. salivarius* 24SMBc was phenotypically characterized as an oral probiotic because of its antagonist activity against pathogenic streptococci. In this study, our aim was to determine the full characterization of *S. salivarius* 24SMBc by whole-genome sequencing (WGS) to gain insights into its antimicrobial activity, considering that genomic information is an excellent source to identify potential natural bioactive products and their biosynthetic pathways. Together with the genome analysis, in vitro assays were carried out to assess the mechanisms of competitive adhesion and co-aggregation of 24SMBc against *S. pneumoniae* and *S. pyogenes* as the main streptococcal pathogens in URTIs. First of all, the 24SMBc genome analysis further confirmed that it is free of streptococcal virulence determinants and, for the first time, we found four different bacteriocin genes: two of them belonging to the IIc class including two, *blpK* and *blpU,* related to BBCG3 characterized by high specific genetic variability for each strain. The *blp* locus that is closely related to peptides produced by *S. pneumoniae* is known to be important for intraspecies competition in *S. pneumoniae* [41], therefore, the peptide homology shared by both microorganisms may explain the strong inhibitory activity of *S. salivarius* 24SMBc against this pathogen and thus its crucial role in interspecies competition within the nasopharynx [12,13,14,15,16,17]. Interestingly, *blpU* and *blpK* were both found in one cluster (BBGC3) that has never been characterized before in the *S. salivarius* species, which usually carries clusters for the biosynthesis of salivaricin 9, salivaricin D, and salivaricin A/B, the most common bacteriocins in this species [20]. On the other hand, the bacteriocin biosynthetic gene clusters (BBGCs) for cerein class IIb bacteriocins, based on our silico analysis, were not found. One possible explanation could be that the current BGC identification tools including those used in this study are largely based on the structure of the BGC database, consequently, BGCs that encode metabolites with previously unrecognized functions may not be available. The output from antiSMASH identified a BGC for type III polyketide synthetase, which was found to be similar to the viguiepinol biosynthetic gene cluster from *Streptomyces* sp. KO-3988 and further characterized in a natural polyketide-isoprenoid hybrid compound [42]. Type III PKSs are self-contained multifunctional enzymes localized at the core of each pathway that convert metabolically-available acyl-CoA precursors into polyketide backbone complexes through a step-by-step chain building mechanism, producing a wide array of natural polyketide compounds, which are widely used in clinical medicines mainly as antibiotics, anti-cancer, and anti-cholesterol compounds [43]. Several studies have reported a convergence between the mevalonate pathway for isoprene biosynthesis from primary metabolism and polyketide biosynthesis from the secondary metabolism also because they share simple precursors such as acetyl-CoA and malonyl-CoA units [44]. This link between primary and secondary metabolism could generate such chimeric assembly lines that highlight biochemical flexibility. As we can understand from what has been said thus far, we could also hypothesize a relationship between primary and secondary metabolisms for *S. salivarius*, considering that it possesses both the Type III PKS cluster and one involved in the biosynthesis of FASII fatty acids, isoprenoid branch mevalonate and a branch of the mevalonate biosynthesis of isoprenoid biosynthesis, and the biosynthesis of polypropylene diphosphate as described by the RAST server. However, further studies focused on the isolation and characterization of the polyketide-isoprenoid compound are planned. The bioinformatic analysis of the genome of 24SMBc also identified 24 different putative adhesins and/or aggregation factors. Streptococci possess an array of surface-associated proteins that interact with eukaryotic cells and the extracellular matrix (ECM), among which there are adhesins essential for the colonization of the human host [45]. Several streptococcal species can effectively colonize the oral cavity and the nasopharynx, but substantial phenotypic heterogeneity exists between different strains, affecting their adhesion, colonization, and biofilm-formation abilities [46]. *S. salivarius* is one of the first colonizers of the oral cavity and URT, is one of the streptococcal species that displays the largest number of both extracellular components and glycosyltransferases, is able to bind epithelial cells, host-derived extracellular molecules, and a number of early, middle and late colonizers, emphasizing its role for microbial community establishment and development [47]. Within the genome of *S. salivarius* 24SMBc, we found several genes involved in adhesion and aggregation including two sequences encoding the antigen I/II family domain and another antigen I/II C2 terminal domain, which are normally expressed by streptococci found in the oral cavity and are important in both processes as they interact with the host and other bacteria [46]. Other adhesion-related sequences where we found encoded adhesin domains belong to the MSCRAMM family (microbial surface components recognizing adhesive matrix molecules), usually linked to the peptidoglycan surface with a sortase-dependent mechanism [48] such as FctA and collagen binding domains that could partially explain the potential adherence competition to human cells. Furthermore, other MSCRAMM domains were found; interestingly, one domain belonged to adhesin SdrC, which is involved in both adhesion and aggregation, and two SdrD adhesin domains that are involved in the adherence to nasal epithelium, along with domains belonging to ClfA and ClfB adhesin families that are shared by other Gram-positive cocci with similar tropism [48]. Other domains of interest include mucin-binding ones such as MucBP_2 and Mub B2 for strong adhesion on the mucus, as recently demonstrated in *S. salivarius* F6-1 in which their gene knock out reduced adherence on the HT29-MTX cells; whereas for aggregation, glucan-binding GbpC domain played a role in dextran-dependent aggregation in other streptococci [40,49], as shown in *S. gordonii*, which is able to co-aggregate with *Actinomyces naueslundii* and *Porphyromonas gingivalis,* and the surface adhesin CshA is also essential for auto-aggregation in *S. salivarius* as previously described [50]. These data matched our results obtained from 24SMBc antagonistic activity against the *S. pneumoniae* and *S. pyogenes* strains of different serotypes. First, 24SMBc showed no cytotoxic effect on the HEp-2 cell line, an extraordinary ability to adhere to the host epithelium, and has a very high value of auto-aggregation (81.8%), which justifies its permanence and predominance in the respiratory microbiota in clinical studies. Since nasopharynx colonization is the critical step for these streptococcal strains and colonization can lead to infection, competitive adhesion and exclusion are essential preventive features that a probiotic strain can perform in lieu of the host microbiota [51]. 24SMBc significantly reduces pneumococcal adhesion on Hep-2 cells, whereas it does not show the same effects on *S. pyogenes*. It significantly co-aggregates with pneumococci and *S. pyogenes*, even if the co-aggregation with the latter is not significative. In the adhesion interference assay, *S. salivarius* 24SMBc caused a significative reduction of adhering pneumococci strains. The mechanism behind this phenomenon is not yet well known, but this reduction could be due to the competition of binding sites and to the mutual exclusion by co-aggregation. In fact, *S. salivarius* 24SMBc not only determined a reduction in the pneumococci strains but also showed a good and significative co-aggregation with them. Conversely, the results obtained for *S. pyogenes* showed no significant reduction in adhesion in the presence of *S. salivarius* 24SMBc and no significative co-aggregation. It is likely that the previously described *S. salivarius* 24SMBc inhibitory activity could be linked to the production of bacteriocins and not used for the competition of binding sites. Although adhesion is the pivotal step for colonization, aggregation factors are essential for auto-aggregation and co-aggregation to incorporate other microorganisms and create complex communities on the mucosa [46], a goal that can be essential when using probiotics during or after antibiotic treatment to replenish the local microbiota. Moreover, the aggregation of commensal strains appeared to be necessary for the adhesion to epithelial cells and to enable the formation of a barrier that protects the host’s epithelium from colonization by pathogens, and the ability to co-aggregate with a pathogen allows probiotics to entrap it. These two features are considered as desirable properties for probiotic strains, in fact, the auto-aggregation ability test, together with co-aggregation, could be used as a preliminary screening, identifying potentially adherent bacteria with properties suitable for commercial purposes. In conclusion, our analysis demonstrated, for the first time, three different biosynthetic gene clusters related to the synthesis of bacteriocins and polyketides by genome mining. Likewise, several putative adhesion and aggregation factors were identified on streptococci that could support 24SMBc anti-adhesion and co-aggregation effects, however, further studies and experimental validations will be needed. Our findings further validate the value of the genomic approach to explore the full potentiality of streptococci to produce bioactive natural products for developing new applications for health promotion. This is particularly true in this era, in which the alarming and global impact of the antimicrobial resistance in medicine, in agriculture and animal husbandry has been observed [3]. The use of commensal microorganisms producing natural “competitors” can be used as an alternative strategy that might mitigate this current medical crisis, coupling antibiotic therapy with adjuvant strategies to treat infections.

## Figures and Tables

**Figure 1 microorganisms-10-02042-f001:**
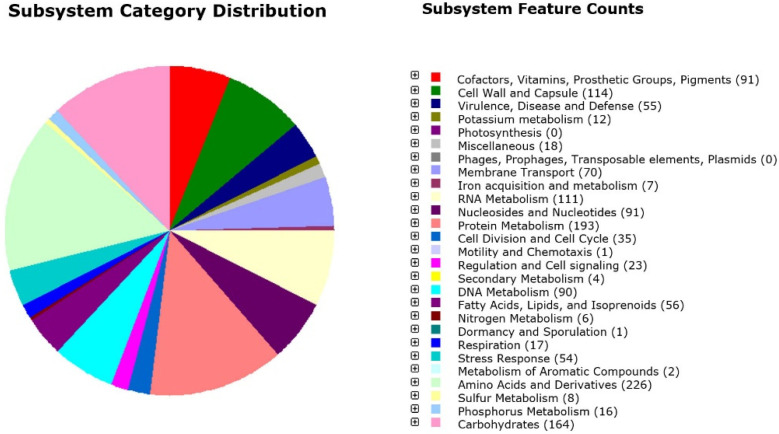
Subsystem category distribution of *Streptococcus salivarius* 24SMBc revealed by genome annotations based on the RAST server. Each color represents a subsystem category with the feature counts listed on the right.

**Figure 2 microorganisms-10-02042-f002:**
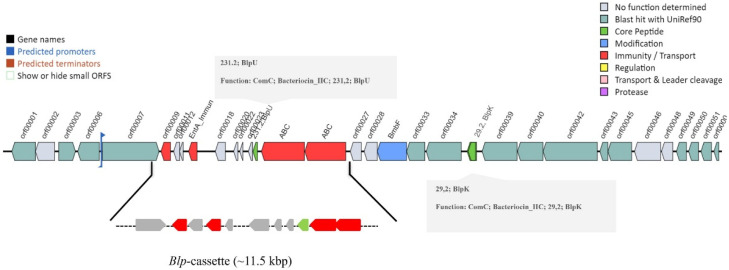
Genetic arrangement of the *blp* locus in *S. salivarius* 24SMBc produced with the BAGEL4 tool. Interestingly, the presence of the other *blpK* bacteriocin is to the right of the *blp* cassette carrying *blpU*.

**Figure 3 microorganisms-10-02042-f003:**
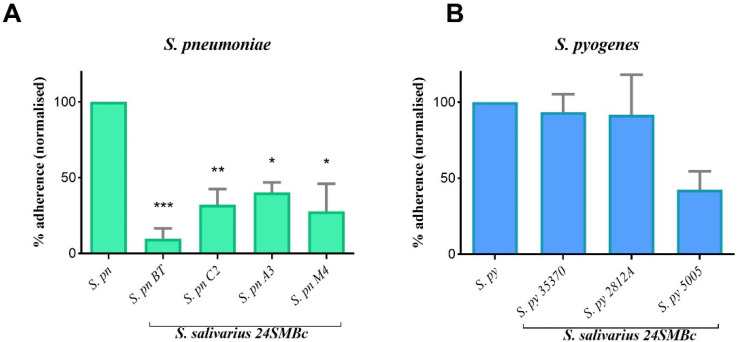
Antagonistic effect of *S. salivarius* 24SMBc against streptococcal pathogens. (**A**) Adherence percentage of *S. pneumoniae* BT (*S. pn* BT), *S. pneumoniae* C2 (*S. pn* C2), S. *pneumoniae* A3 (*S. pn* A3), and *S. pneumoniae* M4 (S. *pn* M4) normalized to *S. pneumoniae* alone (*S. pn*). (**B**) Adherence percentage of *S. pyogenes* Spy3537 (*S. py* 35370), *S. pyogenes* 2812A (*S. py* 2812A), and *S. pyogenes* MGAS 5005 (*S. py* 5005), normalized to *S. pyogenes* alone (*S. py*). Statistically significant *p* value: * *p* < 0.05, ** *p* < 0.01, and *** *p* < 0.001.

**Figure 4 microorganisms-10-02042-f004:**
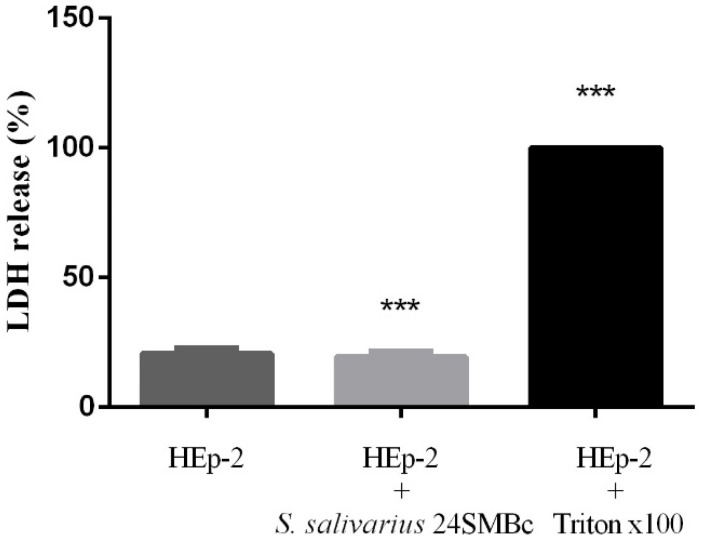
Levels of LDH release to evaluate the cytotoxic effect with *S. salivarius* 24SMBc. HEp-2 alone as 0% LDH release, HEp-2 plus *S. salivarius* 24SMBc, and HEp-2 cells plus Triton X-100 as a control of total release (100% LDH release). Statistically significant *p* value: *** *p* < 0.001.

**Figure 5 microorganisms-10-02042-f005:**
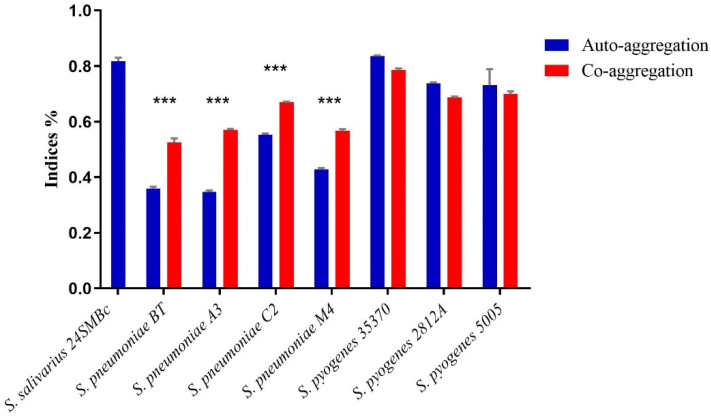
Auto-aggregation and co-aggregation tests. Values of auto-aggregation in blue and co-aggregation in red showed the behavior of each strain when in the presence of *S. salivarius* 24SMBc. Statistically significant *p* value: *** *p* < 0.001.

**Table 1 microorganisms-10-02042-t001:** Main features of the *Streptococcus salivarius* 24SMBc genome. N50 (value to assess the contiguity of assembly, by the length of the shortest contig at 50% of the total size of the assembly); L50 (the number of contigs at 50% of the total size of the assembly); CDS (coding sequence).

Genomic Features	
Size (bp)	2,131,204
G + C content (%)	39.85
N50 length (bp)	171,173
L50	5
No. of contigs	32
Number of Subsystems	217
CDS	1.933
rRNA	6
tRNA	44

**Table 2 microorganisms-10-02042-t002:** Bacteriocins detected in the *S. salivarius* 24SMBc genome: property, amino acid sequence, and their localization in the genome.

Bacteriocin	Property	Amino Acid Sequences	CDS	Genome Locus Tag
*blpU*-like	Class IIc bacteriocin with double-glycine leader peptide (PF10439)	MTTQTMNNFETLDLEALANVEGGGWVKCYAGTIGSALVGSAGGPVGYWGGALVGYATFC	Contig_24_87554_87375	LFGOLEEL_01643
*blpK*-like	Class IIc bacteriocin with double-glycine leader peptide (PF10439)	MTTQIINNFNSLNSEDLSIIEGGGVIGCVAGTAGSAGLGFLTGTSVGTVTFPIVGTVSGGAFGA	Contig_24_96179_95985	LFGOLEEL_01651
bact_IIb_cerein family	Class IIb bacteriocin, lactobin A/cerein 7B family (TIGR03949)	MTNKERNTSDLASVTGGGWKTNLAVGGLCLASGPIGTMICLGAYNGYMDSAR	Contig_26_37678_37836	LFGOLEEL_01810
bact_IIb_cerein family	Class IIb bacteriocin, lactobin A/cerein 7B family (TIGR03949)	MTKTINNRKHMTTQELEAVSGGVVPWAAISVGMAAAKLTYDLSYAAGKSFYNLTH	Contig_26_37857_38024	LFGOLEEL_01811

**Table 3 microorganisms-10-02042-t003:** Putative proteins involved in the aggregation and adhesion of *S. salivarius* 24SMBc.

Genome Locus Tag	Protein/ Domain	Property	Putative Function
LFGOLEEL_00404	GbpC superfamily	Streptococcus glucan-binding protein C (PF08363)	Adhesion, dextran-induced aggregation
	Streccoc_I_II	Antigen I/II family LPXTG-anchored adhesin (NF033804)	Adhesion, aggregation
LFGOLEEL_00415	GBS Bsp-like	GBS Bsp-like repeat (PF08481)	Adhesion
	SH3_5	SH3 domain-containing protein (PF08460)	Adhesion, aggregation
LFGOLEEL_00538	Collagen_binding domain	Collagen_bind (PF05737)	Adhesion, colonization
	FctA	FctA family (PF12892)	Adhesion, colonization
LFGOLEEL_00802	Agg_substance superfamily	LPXTG-anchored aggregation substance (NF033875)	Aggregation
LFGOLEEL_00836	CshA_NR2	Surface adhesin CshA non-repetitive domain 2 (PF18651)	Adhesion, aggregation
	CshA_repeat	Surface adhesin CshA repetitive domain (PF19076)	Adhesion
	CshA_fibril_rpt	CshA-type fibril repeat (TIGR04225)	Adhesion
	SSSPR-51	SSSPR-51 domain (PF18877)	Adhesion
LFGOLEEL_00906	Streccoc_I_II	Antigen I/II family LPXTG-anchored adhesion (NF033804)	Adhesion, aggregation
	LPXTG anchor	LPXTG-motif cell wall anchor domain (TIGR01167)	Adhesion
LFGOLEEL_00907	GbpC superfamily	Streptococcus glucan-binding protein C (PF08363)	Adhesion, dextran-induced aggregation
	ProTailRpt superfamily	Proline-rich tail region repeat (TIGR04307)	Adhesion
LFGOLEEL_01308	MucBP_2	Mucin binding domain (PF17965)	Mucin adhesion
	Mub_B2	Mub B2-like domain (PF17966)	Mucin adhesion
LFGOLEEL_01314	MucBP_2	Mucin binding domain (PF17965)	Mucin adhesion
	Mub_B2	Mub B2-like domain (PF17966)	Mucin adhesion
LFGOLEEL_01317	MSCRAMM_ClfA	MSCRAMM family adhesin clumping factor ClfA (NF033609)	Adhesion
LFGOLEEL_01318	MSCRAMM_SdrD	MSCRAMM family adhesin SdrD (NF012181)	Adhesion
	aRib	Atypical Rib domain (PF18938)	Adhesion
LFGOLEEL_01320	MSCRAMM_SdrC	MSCRAMM family adhesin SdrC (NF000535)	Adhesion, aggregation
	MSCRAMM_SdrD	MSCRAMM family adhesin SdrD (NF012181)	Adhesion
	MSCRAMM_ClfB	MSCRAMM family adhesin clumping factor ClfB (NF033845)	Adhesion, colonization
LFGOLEEL_01321	Agg_substance super family	LPXTG-anchored aggregation substance (NF033875)	Aggregation
LFGOLEEL_01338	MucBP_2	Mucin binding domain (PF17965)	Mucin adhesion
	Adhesin_LEA super family	LEA family epithelial adhesin N-terminal domain (NF033647)	Adhesion
LFGOLEEL_01344	AgI_II_C2	Cell surface antigen I/II C2 terminal domain (PF17998)	Adhesion, aggregation
	Agg_substance super family	LPXTG-anchored aggregation substance (NF033875))	Aggregation
	FctA	Spy0128-like isopeptide containing domain (PF12892)	Adhesion
LFGOLEEL_01933	DUF4097 superfamily	Putative adhesin (cl23960)	Adhesion

## Data Availability

The *S. salivarius* 24SMBc genomic reads were deposited in the National Center for Biotechnology Information (NCBI) SRA database under SRR20769826, the BioProject accession number PRJNA865118.

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
