# Peer review of "Streptococcus salivarius 24SMBc Genome Analysis Reveals New Biosynthetic Gene Clusters Involved in Antimicrobial Effects on Streptococcus pneumoniae and Streptococcus pyogenes"

_microorganisms, 2022, doi:10.3390/microorganisms10102042_

Round 1
Reviewer 1 Report
The manuscript “Streptococcus salivarius 24SMBc Genome Analysis Reveals New Biosynthetic Gene Clusters through Genome Mining Involved in Antimicrobial Effects on S. pneumoniae and S. pyogenes Strains” presents the results the authors got performing and analyzing the genome sequence of Streptococcus salivarius 24SMBc, an oral probiotic with antimicrobial activity against the otopathogens Streptococcus pyogenes and Streptococcus pneumonia.
The authors sequenced Streptococcus salivarius 24SMBc genome and through Genome Mining found interesting biosynthetic gene clusters, that are well described.
The manuscript is of interest, the results are well presented and the methods well described. Thus, in my opinion, the manuscript deserves to be published after minor revisions.
The introduction is comprehensive of all the necessary information. Methods are detailed and results are corroborated by statistical analysis.
Minor revisions:
The authors should
-add the explanation to the L50, CDS in the table;
-explain what they mean with subsystems;
-check the style of Fig 1 since red lines are visible
-remove one signature, pg 6 “Fifteen of them displayed a typical cell- wall signature signature with LPxTG motif”
-replace the sentence “for each S. salivarius strains “with for each S. salivarius strain”, pg 8
-use italics to indicate bacterial names, i.e. Streptomyces pg 8 and 11
-correct resulting not statically significant, pg 11
-check the sentence “First of all, the 24SMBc genome analysis further confirmed that it is free of streptococcal virulence determinants and, for the first time, four different bacteriocin genes ; were detected: two of them belonging to the IIc class including two, blpK and blpU, related to BBCG3 characterized by high specific genetic variability for each strain..
Reviewer 2 Report
In this work, the full characterization of Streptococcus salivarius 24SMBc was obtained by whole-genome sequencing and annotation, and showed anti-adhesion and co-aggregation effects against S. pneumoniae and S. pyogenes strains in vitro assays, which confirmed that 24SMBc had antimicrobial activity against pathogenic streptococci. It is significant to understand the antimicrobial function of Streptococcus salivarius 24SMBc. However, some issues exist in the manuscript.
1 The title of the manuscript is too long, please modified it.
2. There are some logical problems in Abstract part, please readjust and strengthen the novelty of this study.
3. Page 2. “to interfere with streptococcal pathogens and to carry bacteriocins such as blp-U in the blp-U like cassette as previously described by Santagati et al.” remove to.
4. Page 2. “From the vast pool of α-streptococci, the oral probiotic strain Streptococcus salivarius 24SMBc has been used for local bacteriotherapy, especially against other pathogenic streptococcal strains, namely Streptococcus pyogenes and Streptococcus pneumonia, which are important pathogens in UR-TIs.” Please provide the relevant references. If not, please supplement the relevant experiments.
5. Please show the relationship between Streptococcus salivarius 24SMBc and natural bioactive products.
6. It is advisable to list the differences between the current research and other published research work, and discuss the importance of this research work in introduction.
7. Why is the experimental conclusions displayed at the end of Introduction?
8. Please provide the source of S. salivarius 24SMBc in Materials and Methods.
9. What is the quality test result of DNA extraction? Please show the results of agarose gel electrophoresis. Genomic DNA of 24SMBc was quantified by Qubit 2.0 fluorometer and its quality was assessed by Nanodrop2000 Spectrophotometer. Please show the results of Qubit and Nanodrop, and explain whether the total sample size and concentration meet the requirements in this manuscript.
10. The quality of Figure 2 should be improved.
11. Please provide the auto-aggregation results of Streptococcus salivarius alone in Figure 5 as the control group. In addition, please check whether the percentage of ordinate is correct.
12. Format problems of histogram, bold, font, font size, color, first letter case of coordinate name, please keep consistent.
13. Page 11. The data should be corrected: 81.8 ± 0.012%, 78.6 ± 0.005% and 52.5 ± 0.015%
14. There are many grammar and format issues. For example, the tenses in the same sentence are inconsistent. The authors are suggested to go through the whole manuscript again, and check punctuation, potential typos and grammar problems.
15. Format problems of references, such as title case, font format, please check and modify it.
Author Response
I would like to thank the reviewers for their insightful comments that were important for the paper revision and its improvement. We have revised the manuscript following your comments and we hope this will meet with your approval. The revised portions are marked up by track changes function in the paper as request, below are our point-by-point response to the reviewers’ comments.
Point 1: The title of the manuscript is too long, please modified it
Response 1: We agree with the reviewer, we modified the title
Point 2 There are some logical problems in Abstract part, please readjust and strengthen the novelty of this study.
Response 2: thanks for this comment, we revised the abstract
Point 3. Page 2. “to interfere with streptococcal pathogens and to carry bacteriocins such as blp-U in the blp-U like cassette as previously described by Santagati et al.” remove to.
Response 3: text modified and removed this sentence, pag. 3
Point 4. Page 2. “From the vast pool of α-streptococci, the oral probiotic strain Streptococcus salivarius 24SMBc has been used for local bacteriotherapy, especially against other pathogenic streptococcal strains, namely Streptococcus pyogenes and Streptococcus pneumonia, which are important pathogens in UR-TIs.” Please provide the relevant references. If not, please supplement the relevant experiments.
Response 4: Sorry for the mistake, we corrected the reference in the text, pag 2, reference 10
Point 5. Please show the relationship between Streptococcus salivarius 24SMBc and natural bioactive products.
Response 5: thanks for this comment, we revised the text to better explain the relationship between S. salivarius and NP, pag.3. In particular, S. salivarius 24SMB is a bacteriocin producer i.e. blp bacteriocin, that is involved in antimicrobial activity. The bacteriocin i.e. ribosomal synthesized and post-translationally modified peptides (RiPP) together non-ribosomal peptides (NRP) and polyketides (PK) are natural bioactive products (NPs) usually with antimicrobial activity, regulated by biosynthetic gene clusters (BGCs) on the genome.
Point 6 It is advisable to list the differences between the current research and other published research work, and discuss the importance of this research work in introduction.
Response 6: we added the text in the introduction, pag. 2. However, to date two oral probiotic have been well characterized and commercially available, S. salivarius K12 and M18 producers of the bacteriocin sal A/B with antimicrobial activity mainly against S.pyogenes and S. salivarius 24SMBc and S.oralis, the former produces the bacteriocin blp with antimicrobial activity against S.pyogenes and mainly S.pneumoniae.
Point 7 Why is the experimental conclusions displayed at the end of Introduction?
Response 7: We agree with the reviewer for this comment, we deleted this part of the text
Point 8. Please provide the source of S. salivarius 24SMBc in Materials and Methods.
Response 8: we added the reference in the text. However, S. salivarius 24SMBc isolated from nasal swabs of healthy children and was selected for its remarkable ability to interfere with URTIs, mainly AOM pathogens i.e. S. pyogenes and S. pneumoniae
Point 9. What is the quality test result of DNA extraction? Please show the results of agarose gel electrophoresis. Genomic DNA of 24SMBc was quantified by Qubit 2.0 fluorometer and its quality was assessed by Nanodrop2000 Spectrophotometer. Please show the results of Qubit and Nanodrop, and explain whether the total sample size and concentration meet the requirements in this manuscript.
Point 9. Sorry, we meant DNA purity, we modified this part in the text. However, the genomic DNA extracted was measured using Qubit dsDNA obtaining the concentration of 1.4 µg/µl, while we determined the purity by Nanodrop2000 evaluating the absorbance ratio 260/280 and 260/230 as indicator of protein contamination and organic compounds. DNA extraction showed the value of 1.9 and 2.15 for 260/280 and 260/230. respectively, then it appears to be as a pure DNA and suitable for sequencing use. We added the figure of agarose gel electrophoresis as supplement materials: Figure S1.
Point 10 The quality of Figure 2 should be improved.
Point 10 figure 2 revised
Point 11 Please provide the auto-aggregation results of Streptococcus salivarius alone in Figure 5 as the control group. In addition, please check whether the percentage of ordinate is correct.
Point 11 figure 5 changed and S. salivarius added alone
Point 12. Format problems of histogram, bold, font, font size, color, first letter case of coordinate name, please keep consistent.
Point 12: we agree with the reviewer, we revised figures
Point 13. Page 11. The data should be corrected: 81.8 ± 0.012%, 78.6 ± 0.005% and 52.5 ± 0.015%
Point 13: sorry for the mistake, we corrected the text
Point 14. There are many grammar and format issues. For example, the tenses in the same sentence are inconsistent. The authors are suggested to go through the whole manuscript again, and check punctuation, potential typos and grammar problems.
Point 14: thank for this comment, the work was fully double-checked for English language
Point 15 Format problems of references, such as title case, font format, please check and modify it.
Point 13: references revised

Round 2
Reviewer 2 Report
1. Please keep the names “Streptococcus pyogenes/Streptococcus pneumoniae” described consistent in Abstract part.
2. “Streptococcus salivarius 24SMBc is an oral probiotic with antimicrobial activity against the otopathogens Streptococcus pyogenes and Streptococcus pneumoniae. Clinical studies have reinforced its role in reducing the recurrence of upper respiratory tract infections (URTIs) and rebalancing the nasal microbiota.” The antimicrobial activity of Streptococcus salivarius 24SMBc reported in other literatures is introduced at the beginning of this manuscript, which is consistent with the antimicrobial result described in this study. Please highlight the related advantage of this study.
3. In 2.2, the unit “u/l” should be corrected to “u/L”.
Author Response
I would like to further thank the reviewer for his comments to improve the manuscript. We have revised the text following his comments and hope this will meet with your approval. Revised portions are marked with the change tracking feature in the document as required, below are our point-by-point responses to the reviewer’s comments.
Point 1: Please keep the names “Streptococcus pyogenes/Streptococcus pneumoniae” described consistent in Abstract part.
Response 1: text revised
Point 2. “Streptococcus salivarius 24SMBc is an oral probiotic with antimicrobial activity against the otopathogens Streptococcus pyogenes and Streptococcus pneumoniae. Clinical studies have reinforced its role in reducing the recurrence of upper respiratory tract infections (URTIs) and rebalancing the nasal microbiota.” The antimicrobial activity of Streptococcus salivarius 24SMBc reported in other literatures is introduced at the beginning of this manuscript, which is consistent with the antimicrobial result described in this study. Please highlight the related advantage of this study.
Response 2: we modified the abstract. However, in this study we determined the whole genome sequencing it in combination with antiSMASH and BAGEL to investigate the bacterocion and metabolite productions to better understand 24 SMBc antibacterical activity. Interestingly, in this study we found that 24 SMBc posseses four bacterions linked to anti streptococcal activity and one BGC related to Type 3PKS that could have beneficial effects as reported in the literature for this componds. Further, we evaluated the antimicrobial effects in terms of aggregation ability and competitiveness between 24SMbc and pathogens that had never been tested before
Point 3. In 2.2, the unit “u/l” should be corrected to “u/L”.
Response 3: text corrected
